# Comparison of Hot Deformation Behavior Characteristics Between As-Cast and Extruded Al-Zn-Mg-Cu (7075) Aluminum Alloys with a Similar Grain Size

**DOI:** 10.3390/ma12233807

**Published:** 2019-11-20

**Authors:** H.T. Jeong, W.J. Kim

**Affiliations:** Department of Materials Science and Engineering, Hongik University, Mapo-gu, Sangsu-dong 72–1, Seoul 121–791, Korea; heerae1051@gmail.com

**Keywords:** aluminum alloys, casting, extrusion, grain size, segregated phases, hot compression, processing maps

## Abstract

The hot compressive behavior and processing maps of as-cast and extruded 7075 aluminum alloys with a similar grain size (320–350 μm) were studied and compared, which allows us to directly observe the effect of segregated phases in the as-cast microstructure on the deformation behavior and hot workability of 7075 alloys. In the as-cast alloy, the compound phases segregated along the interdendritic interfaces within the interiors of original grains provided the additional sites for continuous dynamic recrystallization via the particle stimulation nucleation mechanism. As a result, the as-cast alloy exhibited higher fractions of recrystallized grains and smaller grain sizes than the extruded alloy after compression. The stress exponent values of the as-cast alloy were smaller than those of the extruded alloy. In the processing maps, the domain associated with high power dissipation efficiencies (≥35%) occurred in a wider temperature range in the as-cast alloy compared to the extruded alloy. The segregated phases that remained undissolved in the as-cast alloy after compressive deformation could be effectively eliminated during the solid solution treatment (753 K for 2 h) for T6 aging applied after hot compression. The current results suggest the possibility and advantage of omitting the extrusion step when preparing 7xxx aluminum forging or extrusion feedstocks for hot working. The proposed method can be applied to other precipitation hardenable aluminum alloys.

## 1. Introduction

The Al-Zn-Mg-Cu aluminum alloy system (7xxx series) has a high strength-to-density ratio and good toughness coupled with good resistance to stress corrosion cracking [1,2,3,4]. Due to these properties, it has been widely used in the aerospace and automobile industries for heavy structural applications [5,6,7]. Forging and extrusion methods have been utilized to manufacture various 7xxx aluminum alloy components [8,9,10] and to determine the optimal hot forming process conditions for these methods, high-temperature compressive deformation behavior and processing maps of 7xxx alloys have been studied [11,12,13,14,15,16,17,18,19,20,21].

Yang et al. [14] studied the hot deformation behavior and processing maps of an extruded 7075 alloy in the temperature range of 593–753 K and in the strain rate range of 10^−3^–1 s^−1^. The optimum hot working condition was in the temperature range of 673–753 K and in the strain rate range of 10^−2^–10^−1^ s^−1^. Dynamic recovery (DRV) occurred in the stable regions, and the activation energy for plastic flow was close to that of self-diffusion energy in pure aluminum. Yang et al. [15] also studied the effect of initial grain size on the processing maps of the extruded 7075 alloy. The power dissipation efficiency in the safe region was higher in the fine-grained alloy than that for alloys with coarse grain sizes at the same conditions. Rokni et al. [16] examined the microstructural evolution of an extruded 7075 alloy after hot compressive deformation in the temperature range of 723–853 K. Continuous dynamic recrystallization (CDRX) occurred during hot compression. Sun et al. [17] examined the microstructural evolution of an extruded 7075 alloy after hot compressive deformation in the temperature range of 573–723 K. At temperatures below 623 K, DRV occurred at and above 623 K, partial dynamic recrystallization (via CDRX) occurred. While most of researchers studied the hot deformation or/and processing maps of the extruded 7075 alloys, as reviewed above, some researchers studied the hot deformation characteristics of the as-cast 7075 alloys. Lianggang et al. [18] studied the processing map of the as-cast 7075 alloy in the temperature range of 573–773 K and in the strain rate range of 10^−2^–10 s^−1^. They suggested that the suitable processing region for hot working was in the temperature range of 698–738 K and in the strain rate range of 10^−2^–1 s^−1^, and DRV mainly occurred in that region. Park and Kim [19] studied the difference in the hot compressive behavior and processing map between as-cast and homogenized cast 7075 alloys in the temperature range of 573–723 K and in the strain rate range of 10^−3^–10 s^−1^. The results showed that the as-cast alloy had higher power dissipation efficiencies and smaller regions for unstable flow compared to the homogenized 7075 alloy. This was because the segregated compound phases in the solidified microstructure of the as-cast alloy played an important role in grain refinement by DRX. The same authors also studied the original grain size effect on the processing maps of the as-cast 7075 alloy [20] and showed that the reduction of the grain size by electromagnetic stirring improved the hot workability.

In the present work, we studied the hot deformation behavior and processing maps of the as-cast and extruded 7075 alloys with a similar grain size. The goals were to compare the hot deformation behavior, dynamic recrystallization and hot workability between the as-cast and extruded 7075 alloys under the same grain-size effect and examine the possibility of omitting the extrusion step in preparing 7xxx aluminum forging or extrusion feedstocks for hot working. The deformation mechanisms of the two alloys were discussed, and the optimum conditions for hot working were compared. The microstructures after hot working were also compared.

## 2. Materials and Methods 

An Al-Zn-Mg-Cu alloy (5.93Zn–2.28Mg–0.55Cu–0.11Fe–0.23Si–bal Al by wt.%) ingot with a diameter of 200 mm was fabricated by the direct chill casting method. After the homogenization treatment on the ingot at 753 K for 24 h, direct extrusion was conducted with an extrusion ratio of 12 at 623 K.

Hot compression tests were conducted on the as-cast and extruded alloys at 573, 623, 673 and 723 K with 50 K intervals under the five different strain rates of 10^−3^, 10^−2^, 10^−1^, 1 and 10 s^−1^ using a Gleeble 3500 thermomechanical simulator. The compression specimens had a cylinder shape with a diameter of 10 mm and a height of 12 mm. The compression axis of the samples was parallel to either the extrusion or longitudinal direction of solidification. The heating rate for the specimens to reach the target temperature in the Gleeble was 10 K/min, and the sample holding time was 3 min. After the compressive deformation reached a true strain of 1.2, the test was stopped and the deformed sample was immediately removed from the chamber and water quenched to capture the high-temperature microstructure. A K-type thermocouple was spot welded to the surface of the longitudinal body of the sample to monitor the temperature during the compression test. The adiabatic temperature rise during deformation was recorded using the thermocouple. The true stress–true strain curves were plotted based on the raw data obtained from the compression tests. The values of flow stress were corrected for the adiabatic temperature increase by plotting the linear interpolation between ln *σ* and l/*T*, where *σ* is the flow stress and *T* is the instantaneous temperature.

In identifying the phases in the as-cast and extruded alloys, high-resolution X-ray diffraction (HR-XRD, SmartLab, Tokyo, Japan) was performed using Cu K_α_ radiation (wavelength λ = 0.1541 nm) at 45 kV and 200 mA.

The microstructures of the specimens were examined using optical microscopy (Olympus BX51M, Tokyo, Japan) and scanning electron microscopy (SEM, JSM-7100F, Tokyo, Japan) coupled with energy dispersive X-ray spectroscopy (EDS) and electron backscatter diffraction (EBSD). The microstructure observation was made at a position that was 1/3 away from the surface toward the center of the sample. To prepare the samples for the EBSD analysis, the cross-sectional areas parallel to the compression axis of the specimens were mechanically ground, and the polished samples were mechanically ground using SiC paper and then ion-milled. The scanning step size for EBSD was 0.7 μm. The EBSD data were analyzed using TSL-OIM software, and the data points with a confidence index under 0.1 were removed. The grain tolerance angle was set at 5°. The grain orientation spread (GOS) method was used to determine the fractions and sizes of the dynamically recrystallized grains. The grains with a GOS value ≤2° were considered to be dynamically recrystallized. To prepare the samples for the optical and SEM observations, the cross-sections of the samples were mechanically ground, polished and chemically etched with Keller’s etchant in a solution of 190 mL H_2_O + 5 mL HNO_3_ + 3 mL HCl + 2 mL HF.

## 3. Results

### 3.1. Initial Microstructure

Figure 1a,b shows the inverse pole figure (IPF) maps and the image quality maps (overlaid with grain boundaries) of the as-cast and extruded alloys examined by EBSD, respectively. In the as-cast microstructure, nearly equiaxed grains comprise multiple dendrites with similar orientations. The extruded alloy exhibits elongated grains along the extrusion axis direction. The presence of a high fraction of low-angle grain boundaries (0.81) and a very low fraction of dynamically recrystallized grains (0.03) in the extruded microstructure indicate that DRV mainly occurred during the extrusion process. The grain size of the as-cast alloy (352 μm) was similar to that of the extruded alloy (321 μm). Figure 1c,d shows the inverse pole figures of the as-cast and extruded alloys, respectively. The texture of the extruded alloy consists of a 〈111〉 fiber together with a 〈100〉 fiber, with the fiber axis parallel to the extrusion axis, while the as-cast alloy shows a random-like grain orientation. The 〈111〉 and 〈100〉 duplex fiber texture has been reported to form during the axisymmetric extrusion of aluminum [22].

Figure 2a,b shows the SEM micrographs of the as-cast and extruded alloys. The secondary phases are highly segregated along the dendritic interfaces in the as-cast microstructure, forming a semi-continuous network, while they are sparsely and discretely (with a particle-like shape) dispersed over the matrix in the extruded microstructure. The XRD curves and the EDS analysis for the as-cast and extruded alloys shown in Figure 2c suggest that the phases segregated at the dendritic interfaces and grain boundaries in the as-cast microstructure, and the particles dispersed in the extruded microstructure are MgZn_2_, Mg_2_Si, Al_2_CuMg, Al_7_Cu_2_Fe and Al_2_Mg_3_Zn_3_.

### 3.2. Hot Compression Deformation Behavior

Figure 3a–h shows the true stress–true strain curves of the as-cast and extruded alloys, which were obtained from the series of compression tests performed at various temperatures and strain rates. The flow stresses corrected after considering the temperature increase by adiabatic heating during deformation are marked by open symbols. After the flow-stress correction, it is found that all the curves of the two alloys display steady-state flow stress behavior.

Metals and metallic alloys typically show the following relationships between flow stress, strain rate and temperature in the steady state, depending on the strain rate (or flow stress) range:(1)$$\dot{\varepsilon }=A_{1}\sigma^{n_{1}}\exp\left(-Q_{c}/RT\right),$$
(2)$$\dot{\varepsilon }=A_{2}\exp\left(\beta \sigma \right)\exp\left(-Q_{c}/RT\right),$$
where *A*_1_ and *A*_2_ are the material constants, $\dot{\varepsilon }$ is the strain rate, σ is the flow stress, *n*_1_ is the stress exponent, *β* is the constant, *Q_c_* is the activation energy for plastic flow and *R* is the gas constant. Equation (1) can be applicable at low stresses where power-law creep dominates plastic flow and Equation (2) can be applicable at high stresses where power-law breakdown (PLB) occurs. Garofalo [23] proposed a hyperbolic sine creep equation that can be used to describe the power-law creep and PLB behaviors in a single equation under the assumption that the *Q_c_* value in Equation (1) is the same as that in Equation (2):(3)$$\dot{\varepsilon }=A{\left[\sinh\left(\alpha \sigma \right)\right]}^{n}\exp\left(-Q_{c}/RT\right),$$
where *A* is the material constant, *n* is the stress exponent and *α* (=*β*/*n*_1_) is a fitting parameter. Ideally, *n* = *n*_1_. Equation (1) is plotted in the form of $\log\dot{\varepsilon }-\log\sigma$ for the as-cast alloy at ε= 0.6, and $n_{1}\left(=d\log\dot{\varepsilon }/d\log\sigma \right)$ was determined at each temperature (Figure 4a). Equation (2) is plotted in the form of $\log\dot{\varepsilon }-\sigma$ at ε = 0.6, and $\beta \left(=d\ln\dot{\varepsilon }/d\sigma \right)$ was determined at each temperature (Figure 4b). The averages of the values of $n_{1}$ and *β* measured at different temperatures at ε = 0.6 were 7.0 and 0.0173 MPa^−1^, respectively. Using these two values, α was calculated and it was 0.0165 MPa^−1^. Then, by plotting Equation (3) in the form of $\log\dot{\varepsilon }\;\mathrm{vs}.\;\log\left(\sinh\left(\alpha \sigma \right)\right),\;n\left(=d\log\dot{\varepsilon }/d\log\left(\sinh\left(\alpha \sigma \right)\right)\right)$ at ε = 0.6 was calculated from the slope of the linear curve shown in Figure 4c. The average of the *n* values measured at all the testing temperatures was 4.9. The $Q_{c}$ value at ε = 0.6 was determined by using the intercept values (B) measured at different temperatures in the plot shown in Figure 4c because $B=\log A-Q_{c}/2.3RT$, such that $Q_{c}=-2.3R\left[\frac{dB}{d\left(1/T\right)}\right]$. The $Q_{c}$ value calculated from the slope of the linear curve in the plot of *B* – 1/*T* (Figure 4d) was 166.2 kJ/mol. The same concepts and procedures were applied in determining the $n_{1}$, β, α, *n* and $Q_{c}$ values of the extruded alloy.

The α, *n* and $Q_{c}$ values of the extruded alloy were 0.0152 MPa^−1^, 5.7 and 146.6 kJ/mol, respectively. The $Q_{c}$ values of the as-cast and extruded alloys are similar but slightly larger than the activation energy for lattice diffusion in pure aluminum ($Q_{L}$= 142 kJ/mol [24]). It is worthwhile to note that the *Q*_c_ values of the as-cast and extruded 7075 Al alloy are similar to those measured from the 7075 alloys studied by other investigators (160.3–170 kJ/mol [25,26]). 

To confirm the validity of the calculation of $Q_{c}$ and α and to identify the deformation mechanism, $\log\;\sigma \;\mathrm{and}\;\log\;\sin h\left(\alpha \sigma \right)$ were plotted as a function of log Z, where Z is the Zener–Hollomon parameter $\left(=\dot{\varepsilon }\cdot \exp\left(Q_{c}/RT\right)\right)$. This is shown in Figure 4e,f. For the plots, the averages of the $Q_{c}$ and α of the two alloys were used (α = 0.0163 MPa^−1^ and $Q_{c}$ = 156.4 kJ/mol) because their $Q_{c}$ and α values were similar. The use of the same $Q_{c}$ and α values for the two alloys allows for a direct and convenient comparison of flow-stress level between them at a given strain rate and temperature. Good correlations were observed from both of the plots. In Figure 4e, the *n*_1_ values of the as-cast and extruded alloys were close to 5 at small Z values where power-law creep dominated the plastic flow and then increased to be larger than 7 at large Z values. This result indicates that the rate-controlling deformation mechanisms of the as-cast and extruded alloys are identical and are lattice-diffusion controlled dislocation climb creep at low strain rates and high temperatures and PLB at high strain rates and low temperatures. In the plot of $\log Z\;\mathrm{vs}.\;\log\left(\sin h\left(\alpha \sigma \right)\right)$, shown in Figure 4f, the intercept and slope represent logA and *n*, respectively. The measured *A* and *n* values of the as-cast and extruded alloys were 1.4 × 10^11^ s^−1^ and 4.9 and 6.8 × 10^10^ s^−1^ and 5.5, respectively, at ε = 0.6. This result indicates that when the two alloys are compared at the same *Z* value, the as-cast alloy exhibits a smaller stress exponent and a lower flow stress compared with the extruded alloy.

### 3.3. Processing Maps

A processing map, which is composed of a power dissipation map and a flow instability map, was constructed based on the principles of a dynamic material model. According to Prasad [27], a hot deformation workpiece can be considered a power dissipater. The total power (*P*) consumed during hot deformation consists of two complementary parts: *G* represents the power dissipation through plastic deformation, most of which is converted into heat, and *J* represents the power dissipation through microstructure change, such as dynamic recovery, dynamic recrystallization and damage of the material [27]: (4)$$P=\sigma \dot{\varepsilon }=G+J=\int_{0}^{\dot{\varepsilon }}\sigma d\dot{\varepsilon }+\int_{0}^{\sigma }\dot{\varepsilon }d\sigma .$$

When the material governing equation is assumed to be $\sigma =C\cdot {\dot{\varepsilon }}^{m}$ at high temperatures, where *m* is the strain rate sensitivity exponent (=1/*n*_1_), Murty et al. [28] showed that the efficiency of the power dissipation (η), which is valid even in the case when *m* is not a constant at a given temperature, can be calculated using a numerical method in solving the equation below:(5)$$\eta =2\left(1-\frac{1}{\sigma \cdot \dot{\varepsilon }}\int_{0}^{\dot{\varepsilon }}\sigma \cdot d\dot{\varepsilon }\right).$$

Ziegler [29] considered the stable condition in dynamic materials models, and according to the author, flow instability is related to flow localization, and unstable flow occurs if the differential quotient satisfies the following inequality:(6)$$\frac{\partial D}{\partial \dot{\varepsilon }}<\frac{D}{\dot{\varepsilon }},$$
where *D* is the powder dissipation function.

By putting *D* = *J* in Equation (6), Murty et al. [28] derived a flow instability criterion that is valid for any type of σ−$\dot{\varepsilon }$ curve:(7)$$\frac{\partial J}{\partial \dot{\varepsilon }}=\dot{\varepsilon }\frac{\partial \sigma }{\partial \dot{\varepsilon }}=\sigma \frac{\partial \ln\sigma }{\partial \ln\dot{\varepsilon }}=m\sigma \;\mathrm{and}\;\frac{J}{\dot{\varepsilon }}=\eta \cdot \frac{\sigma }{2}\;\Rightarrow \;\xi =2m-\eta <0.$$.

Figure 5a–f show the processing maps of the as-cast and extruded alloys at ε = 0.2, 0.6 and 1.0 that were constructed based on the method developed by Murty et al. [28]. In this study, polynomial fitting was used to obtain the fitting curves for the determination of the *m* values from the $\log\dot{\varepsilon }-\log\sigma$ curves, which is necessary for determining the η and ξ values. The contour values (black) in the power dissipation maps represent the η values as a percentage, and the contour values (red) in the flow instability maps represent negative ξ values. Prasad [30] reported that typical microstructural evolution mechanisms, such as DRV and DRX, were determined by η values of 20%–30% and 35%–45%, respectively. The domain associated with η ≥ 30% (η ≥ 35%) occurs between 573 and 723 K (640 and 723 K) in the as-cast alloy. As temperature increases, the critical strain rate associated with η = 30% increases from 10^−2^ to >10^−1^ s^−1^. The processing maps of the extruded alloy are basically similar to those of the as-cast alloy. However, the domain associated with η ≥ 30% (η ≥ 35%) is smaller in the extruded alloy, such that it is located in the narrower temperature range between 673 and 723 K (698 and 723 K). The domain associated with η ≥ 30% changes marginally as the level of strain increases in both alloys. It is also noted that the instability regime is located above 10^−1^ s^−1^ at all temperatures and at all strain levels in both alloys. Generally, the mechanisms of flow instabilities are associated with adiabatic shear bands, deformation bands and flow localizations [30]. Therefore, the deformation temperatures and strain rates in the instability regime should be avoided during the practical hot forming process. Based on the processing maps, it could be suggested that the best condition for hot working of the two alloys is at 723 K at strain rates of 10^−3^–10^−1^ s^−1^, where the largest η values are achieved under stable flow condition. 

Figure 6a,b shows the strain rate sensitivity (*m*) maps constructed using the *m* values measured at different strain rates and temperatures at a given strain of 0.6. The flow instability regimes were superimposed on the maps. The followings are observed. First, the *m* values follow the same trend as the *η* values. The as-cast alloy exhibits higher *m* values than the extruded alloy and for each alloy, the higher values of *m* domains shift towards higher temperatures and lower strain rates. Second, most of the data associated with *n*_1_ > 7 (PLB regime) belong to the domain for flow instability, suggesting that unstable flow is likely to occur when PLB governs plastic flow. 

### 3.4. Microstructures After Hot Deformation

Figure 7a–h shows the IPF and grain boundary (GB) maps of the as-cast and extruded alloys at 673 and 723 K at 10^−3^ and 10^−1^ s^−1^. After hot compression, the original grains of the as-cast alloy with equiaxed shape were elongated to be perpendicular to a compression direction, while the original grains of the extruded alloy elongated to the extruded direction was compressed to form nearly equiaxed grains.

The average grain size, DRX grain size, the fraction of dynamically recrystallized grains and the fraction of high-angle grain boundaries (HAGBs) are plotted as a function of the Zener–Hollomon parameter in Figure 8a–d. In the plots, the EBSD data from the as-cast and extruded samples deformed at 573 and 623 K and at 10^−3^ and 10^−1^ s^−1^ (their EBSD images are not shown here) are also included. The average grain sizes of the as-cast alloy are smaller than those of the extruded alloy. The DRX grain sizes of the two alloys tend to decrease as the temperature decreases and strain rate increases (i.e., Z value increases). The difference in DRX grain size between the as-cast and extruded alloys is small at large Z values, but the DRX grain sizes of the as-cast alloy are notably larger than those of the extruded alloy at small Z values. This is because large local strain around the segregated particles in the as-cast alloy increases the deformation energy inside the matrix, which promotes the growth process of DRX grains at high temperatures. The fractions of DRX grains and HAGBs tend to decrease as the Z value increases. The fractions of DRX grains are higher in the as-cast alloy compared to the extruded alloy. The fractions of HAGBs are higher in the as-cast alloy compared to the extruded alloy at small Z values, but slightly lower in the as-cast alloy at high Z values (above 10^11^ s^−1^). The fraction of low-angle grain boundaries (LAGBs) may be higher in the as-cast alloy compared to the extruded alloy at low temperatures due to development of extensive substructure around the segregated phase particles. As temperature increases, however, the fraction of HAGBs increases more rapidly in the as-cast alloy because the number of HAGBs will increase in proportional to the number of LAGBs if LAGBs evolve into HAGBs with progressive increase of misorientation of LAGBs, of which the mechanism will be discussed shortly later. The maximum fractions of DRX grains and HAGBs are less than 0.2 and 0.4 in the as-cast and extruded alloys, respectively, indicating that DRV was dominant during hot deformation in both alloys in the investigated experimental range.

Figure 9a–d shows the magnified IPF and GB maps of the as-cast and extruded samples deformed at 723 K–10^−3^ and 10 s^−1^, and the cumulative misorientation (point to origin misorientation) and local misorientation (point to point) along the vectors marked within grain interiors (indicated by arrows). In both alloys, LAGBs and intermediate-angle grain boundaries developed in grain interiors. Incomplete discontinuous intermediate-angle grain boundaries or high-angle boundary segments were frequently observed within grain interiors, which were connected by low-angle or intermediate-angle grain boundaries. There was also evidence that intermediate-angle boundaries evolved to HAGBs near initial grain boundaries, forming new fine grains along the initial grain boundaries. The changes in misorientation along the vectors in both alloys show that the cumulative misorientation along the vectors exceeds 10–15°, indicating that progressive subgrain rotation led to the occurrence of CDRX in the substructure. 

The insets inFigure 7a–h show the inverse pole figures of the as-cast and extruded alloys. After deformation, the initial textures of the two alloys disappeared, and the <110> texture, which is the uniaxial compression texture in face-centered cubic metals [31], developed in both alloys. As the temperature increased and the strain rate decreased, the <110> texture became weak. This is due to the increase in the fraction of dynamically recrystallized grains.

The optical and SEM micrographs of the as-cast and extruded alloys after compression tests at 723 K–10^−3^ s^−1^ are shown in Figure 10a–d, where stable plastic flow is expected to have occurred according to the processing maps (Figure 5). After compressive deformation of the as-cast microstructure, the semi-network structure of the solute compounds was broken into many fragments. Compared to the microstructure before compression (Figure 2a), the fraction of the segregated phases decreased because they dissolved during hot deformation. Nucleation and growth of (sub)grains around the particles was evident in Figure 10a,b, indicating that the segregated phase provided a high stress concentration for the development of strain-induced substructures, which are potential nucleation sites for CDRX. Therefore, the presence of large segregated particles increased the nucleation site density and sped up the recrystallization process. In the extruded alloy with a small amount of particles, however, this kind of substructure was less frequently developed.

Figure 11a,b shows the optical micrographs of the as-cast and extruded alloys after compression tests at 723 K–10 s^−1^, where unstable plastic flow is expected to have occurred according to the processing maps. Compared with the as-cast microstructure obtained at 723 K–10^−3^ s^−1^, where sufficient time was available for a high extent of dissolution of segregated phases during deformation, a large amount of segregated phases were remained at 723 K–10 s^−1^, where much less time was available for dissolution. Many deformation bands and shear bands were introduced in the grain interior during deformation in the extruded alloy. In the as-cast alloy, on the other hand, intensive plastic deformation was confined to occur in the narrow region near the dendritic interfaces where the secondary phases were highly segregated. The characteristics of these two microstructures agreed with the prediction for flow instability by the processing maps.

The EBSD GB maps and SEM micrographs of the compressed samples (at 723 K–10^−3^ s^−1^) after the solid solution treatment (753 K for 2 h) for T6 aging are shown in Figure 12a,d. During the period of heat treatment operation at 753 K, grain growth occurred from 136 to 250 μm and from 135 to 210 μm in the as-cast and extruded alloys, respectively, without significant change in grain morphology and texture. The segregated phases in the as-cast alloy remained after compressive deformation was greatly eliminated during the heat treatment. As a result, there is little difference in the amount of dispersoid or constituent particles between the extruded and cast alloys. Many of the remaining phases in the as-cast and extruded alloys are mainly made up of undissolved Fe-rich phases, Si-rich phases and Si inclusions. It is known that hard and brittle insoluble intermetallic particles such as Al_7_Cu_2_Fe or Si inclusions of the order of 1–10 μm in size are frequently found in commercial 7xxx alloys [32].

## 4. Discussion

The transformation of LAGBs into HAGBs, which leads to CDRX, has been reported to occur by an increase in the misorientation of LAGBs with the progressive accumulation of dislocations into LAGBs and progressive lattice rotation near the boundaries of the original grains [33,34,35]. Grain boundaries provide the nucleation sites for the CDRX [36] because at the grain boundaries, dislocations accumulate due to dislocation pile-up and dislocation rearrangement by cross slip and dislocation climb is accelerated due to the concentrated stress. Grain boundary sliding at original grains may also cause a rapid transformation of LAGBs into HAGBs by rotating subgrains near the grain boundaries [37,38,39]. For these reasons, a small original grain size promotes CDRX.

In addition to the size of the original grains, the amount of large particles (>1 μm) can affect CDRX [36]. In the present study, the difference in the original grain size between the as-cast and extruded alloys was small, and thus, the role of particles on CDRX was considered to be important. Particle-stimulated nucleation (PSN)-induced dynamic recrystallization can occur when the amount of dislocations that accumulates at particles during deformation is high enough to form DRX nuclei. A number of LAGBs are first formed in the deformation zone around the particles, and these LAGBs evolve to HAGBs as dislocations are continuously trapped in the LAGBs. In the grain interiors of the as-cast alloy, there are many interdendritic interfaces decorated with a high density of segregated compounds. These segregated phase particles provide the favorable sites for PSN-CDRX. Due to this, the fraction of dynamically recrystallized grains was higher in the as-cast alloy compared to that of the extruded alloy.

Texture and grain morphology can be other factors that can affect DRX and hot deformation characteristics. Yang et al. [14] examined these effects on the hot deformation behavior and processing map of an extruded 7075 alloy with the elongated grains by conducting hot compression tests on samples in the temperature ranges of 593−753 K and in the strain rate range of 10^−3^–1 s^−1^ with loading directions that differed from the extrusion direction. The flow stresses of the 0°, 45° and 90° specimens were anisotropic due to the effects of elongated grain microstructure and the crystal textures developed during extrusion, but the main characteristics of the hot deformation behavior, namely, the activation energy for plastic flow, deformation mechanism and processing maps, were independent of the loading direction. This result indicates the weak effect of initial crystallographic texture and anisotropic grain morphology on hot deformation characteristics and hot workability of 7075 alloy. Furthermore, it is worthy to note that the similar deformation texture formed in the present two alloys after hot compression tests despite their difference in the initial texture (Figure 7).

The degree of dynamic recrystallization is higher in the as-cast microstructure than that in the extruded microstructure (Figure 8c). This agrees with that the efficiency of the power dissipation is higher in the as-cast microstructure than in the extruded microstructure (Figure 5). The higher fraction of DRX grains in the as-cast alloy compared to that of the extruded alloy also explains why the flow stress of the as-cast alloy is lower than that of the extruded alloy (Figure 4e,f).

## 5. Conclusions

The hot compressive behavior and processing map of the as-cast and extruded 7075 alloys with a similar grain size were studied, and the obtained results were compared. The major findings are as follows:The as-cast alloy had a higher fraction of dynamically recrystallized grains after compressive deformation. This was because the segregated phases in the grain interiors (along the interdendritic interfaces) promoted PSN-CDRX. In the extruded alloy, CDRX occurred near the original grain boundaries. In both alloys, however, the maximum fractions of dynamically recrystallized grains were less than 0.2, indicating that DRV was dominant during hot deformation.Both alloys showed lattice diffusion-controlled dislocation climb creep at low strain rates and power-law breakdown at high strain rates, but the as-cast alloy was weaker and had smaller stress exponents compared to the extruded alloy, which could be attributed to the creation of higher fractions of DRX grains in the former.A comparison of the processing maps indicated that the domain associated with η ≥ 30% (η ≥ 35%) occurred between 573 and 723 K (640 and 723 K) in the as-cast alloy. In the extruded alloy, it occurred in the narrower temperature range between 673 and 723 K (698 and 723 K). The best condition for hot working in both alloys was at strain rates of 10^−3^–10^−1^ s^−1^ at 723 K.Processing maps and the post-deformation microstructures indicated that the as-cast alloy exhibited a higher hot workability than the extruded alloy, under the condition of the similar grain size.The segregated phases in the as-cast alloy remained after compressive deformation could be effectively eliminated during the solid solution heat treatment for T6 aging.The current results demonstrated the possibility and advantage of omitting the extrusion step when preparing 7xxx aluminum forging or extrusion feedstocks for hot working. This finding can be applied to other precipitation hardenable aluminum alloys.

## Figures and Tables

**Figure 1 materials-12-03807-f001:**
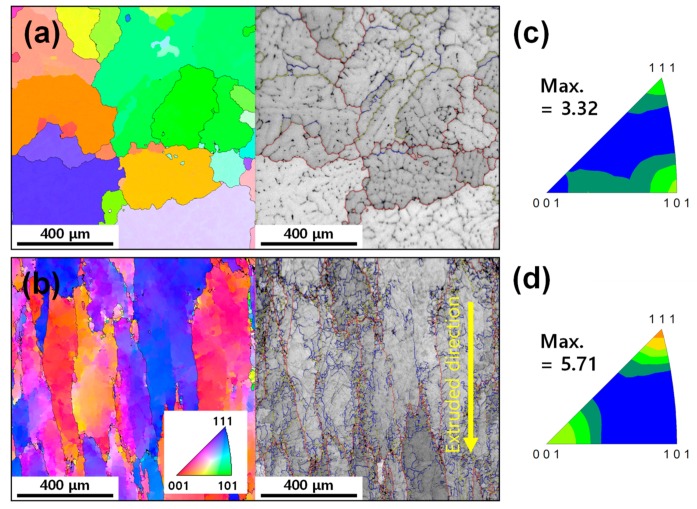
The inverse pole figure (IPF) maps and image quality maps overlaid with grain boundaries of the (**a**) as-cast and (**b**) extruded alloys. The inverse pole figures of the (**c**) as-cast and (**d**) extruded alloys. The maximum texture intensity in the inverse pole figures is in the unit of multiples of a random density (MRD). Low-angle grain boundaries ($2^{{}^{\circ}}\leq \theta <5^{{}^{\circ}}$), intermediate-angle grain boundaries ($5^{{}^{\circ}}\leq \theta <15^{{}^{\circ}}$ ) and high-angle grain boundaries ($15^{{}^{\circ}}\leq \theta$) are represented by blue, yellow and red colors, respectively. The inset in (**b**) shows the color code for the electron backscatter diffraction (EBSD) scans of (**a**) and (**b**).

**Figure 2 materials-12-03807-f002:**
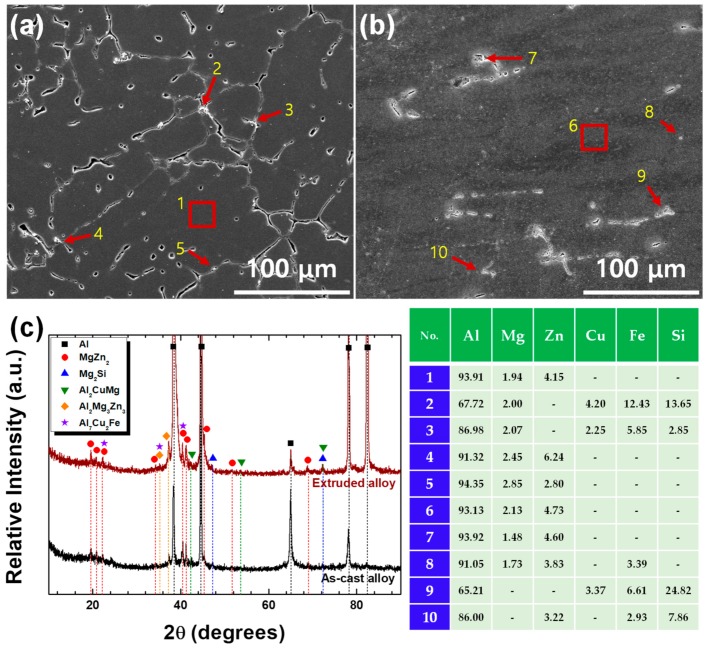
The SEM micrographs of the (**a**) as-cast and (**b**) extruded alloys. (**c**) The XRD curves for the as-cast and extruded alloys. The inserted table represents the energy dispersive X-ray spectroscopy (EDS) analysis results.

**Figure 3 materials-12-03807-f003:**
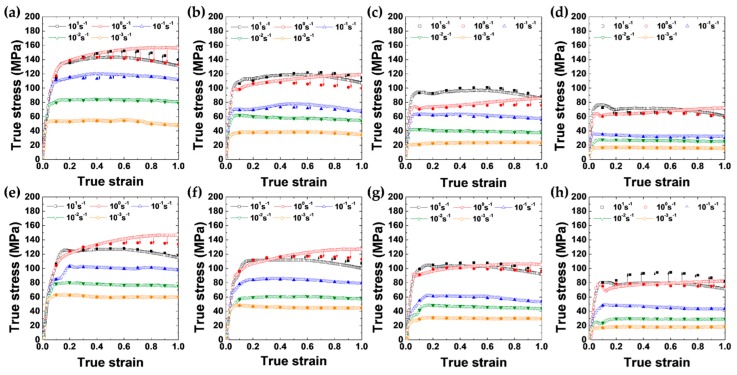
The true stress–true strain curves of the cast alloy at (**a**) 573 K, (**b**) 623 K, (**c**) 673 K and (**d**) 723 K. The true stress–true strain curves of the extruded alloy at (**e**) 573 K, (**f**) 623 K, (**g**) 673 K and (**h**) 723 K. The flow stresses corrected for adiabatic heating are indicated by solid symbols.

**Figure 4 materials-12-03807-f004:**
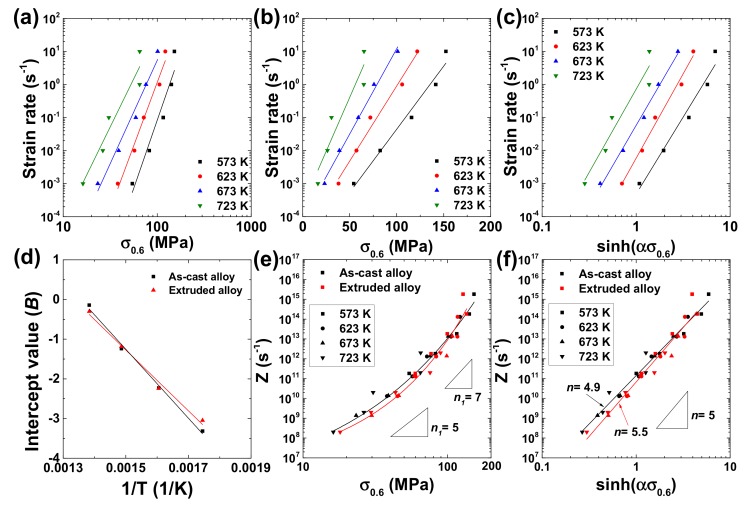
The plots of (**a**) $\log\dot{\varepsilon }-\log\sigma$, (**b**) $\log\dot{\varepsilon }-\sigma$ and (**c**) $\log\dot{\varepsilon }-\;\log\left(\sinh\left(\alpha \sigma \right)\right)$ for the as-cast alloy at ε = 0.6. (**d**) The plots of *B* – 1/*T* for the determination of $Q_{c}$ for the as-cast and extruded alloys at ε = 0.6. The plots of (**e**) log Z−log σ and (**f**) $\log\;Z-\log\;\sin h\left(\alpha \sigma \right)$ for the as-cast and extruded alloys at ε = 0.6.

**Figure 5 materials-12-03807-f005:**
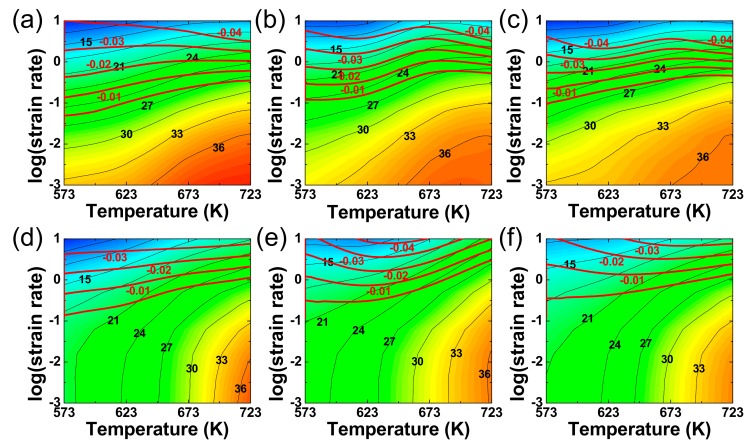
The processing maps of the as-cast alloy at ε values of (**a**) 0.2, (**b**) 0.6 and (**c**) 1.0. The processing maps of the extruded alloy at ε values of (**d**) 0.2, (**e**) 0.6 and (**f**) 1.0. The black color contours and numbers represent the η values, while the reddish color contours and numbers represent the ξ values in the regime of flow instability.

**Figure 6 materials-12-03807-f006:**
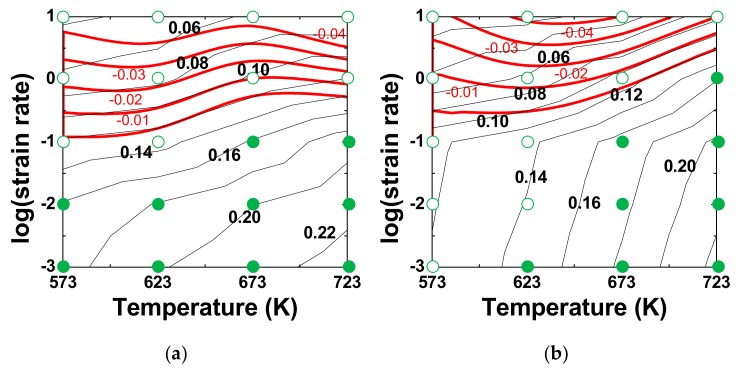
The *m*-maps of (**a**) the as-cast alloy and (**b**) the extruded alloy at ε = 0.6. The iso-efficiency lines of ξ are superimposed on the maps. The black color contours and numbers represent the m values, while the reddish color contours and numbers represent the ξ values in the regime of flow instability. The open symbols represent the experimental conditions belonging to the PLB regime (*n*_1_ > 7), while solid symbols represent the experimental conditions belonging to the power-law creep regime (*n*_1_ ≤ 7).

**Figure 7 materials-12-03807-f007:**
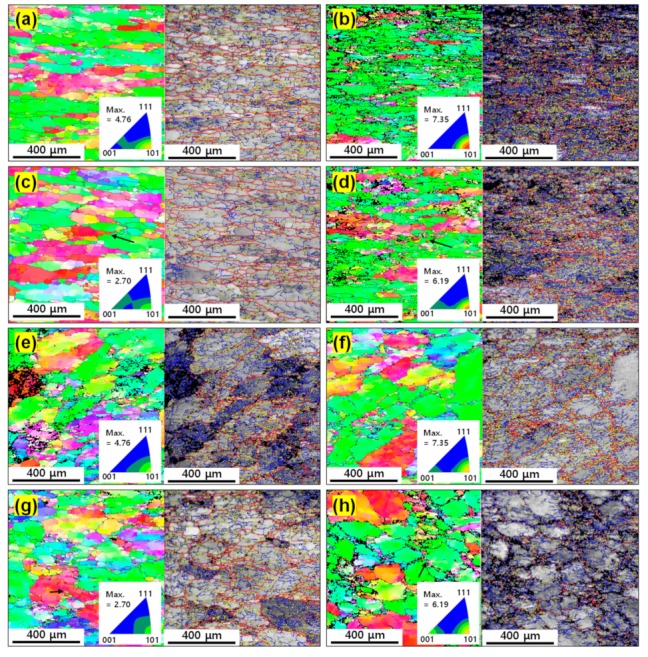
The IPF and grain boundary (GB) maps of the as-cast alloy at (**a**) 673 K–10^−3^ s^−1^, (**b**) 673 K–10^−1^ s^−1^, (**c**) 723 K–10^−3^ s^−1^ and (**d**) 723 K–10^−1^ s^−1^. The IPF and GB maps of the extruded alloy at (**e**) 673 K–10^−3^, (**f**) 673 K–10^−1^ s^−1^, (**g**) 723 K–10^−3^ s^−1^ and (**h**) 723 K–10^−1^ s^−1^. Low-angle grain boundaries ($2^{{}^{\circ}}\leq \theta <5^{{}^{\circ}}$), intermediate-angle grain boundaries ($5^{{}^{\circ}}\leq \theta <15^{{}^{\circ}}$ ) and high-angle grain boundaries ($15^{{}^{\circ}}\leq \theta$) are represented by blue, yellow and red colors, respectively. The insets show the inverse pole figures of the as-cast and extruded alloys.

**Figure 8 materials-12-03807-f008:**
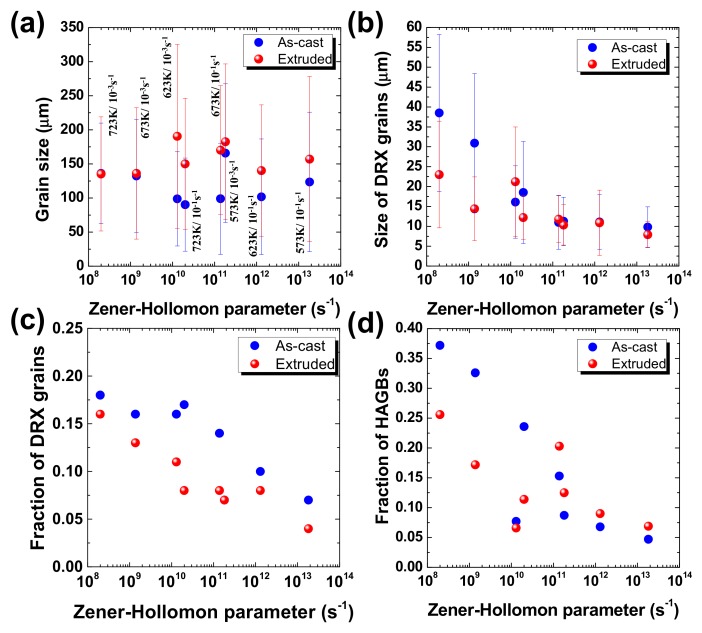
The (**a**) average grain size, (**b**) dynamically recrystallized grain size, (**c**) fraction of dynamically recrystallized grains and (**d**) fraction of high-angle grain boundaries determined from the EBSD analysis, which are plotted as a function of the Zener–Hollomon parameter.

**Figure 9 materials-12-03807-f009:**
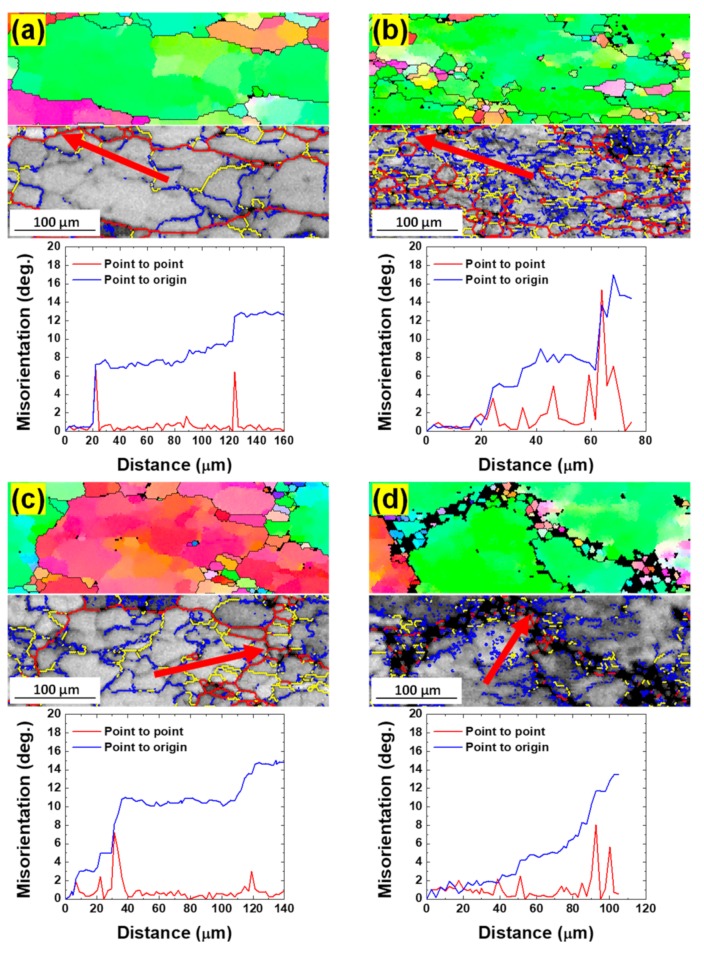
The magnified IPF and GB maps of the (**a**,**b**) as-cast and (**c**,**d**) extruded samples deformed at 723 K–10^−3^ s^−1^ and 723 K–10^−1^ s^−1^ and the cumulative misorientation (point to origin misorientation) and local misorientation (point to point) along the vectors marked on within grain interiors in (**a**–**d**).

**Figure 10 materials-12-03807-f010:**
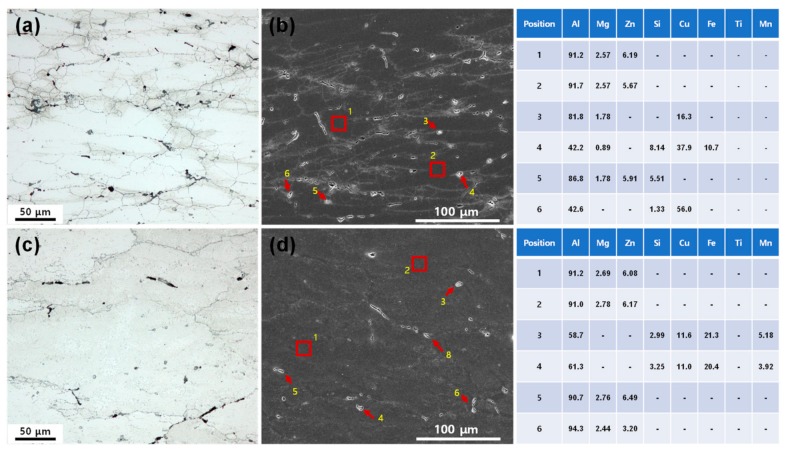
The optical and SEM micrographs of the (**a**,**b**) as-cast and (**c**,**d**) extruded alloys after compression tests at 723 K–10^−3^ s^−1^. The inserted tables represent the EDS analysis results.

**Figure 11 materials-12-03807-f011:**
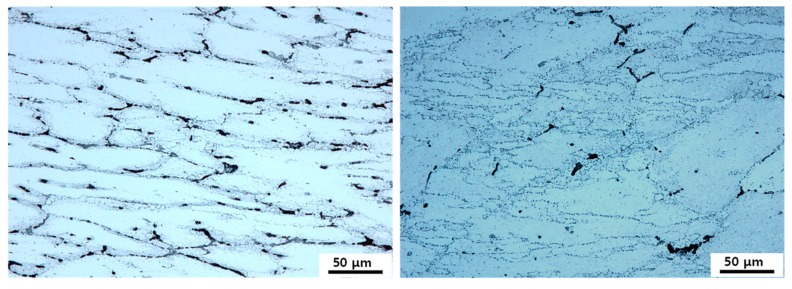
The optical micrographs of (**a**) the as-cast alloy and (**b**) the extruded alloy after compression tests at 723 K–10 s^−1^.

**Figure 12 materials-12-03807-f012:**
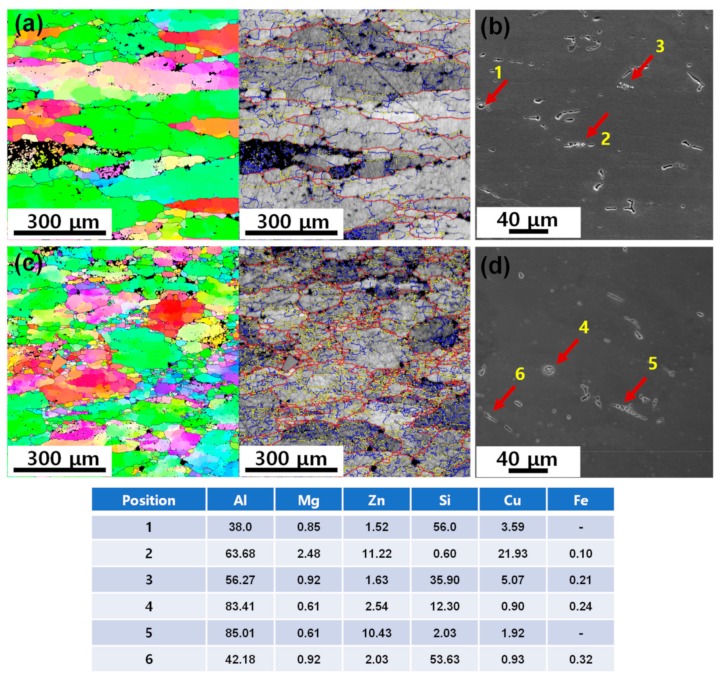
The IPF and GB maps of the (**a**) as-cast and (**c**) extruded alloys after compression tests at 723 K–10^−3^ s^−1^. The SEM micrographs of the (**b**) as-cast and (**d**) extruded alloys after compression tests at 723 K–10^−3^ s^−1^. The inserted table represents the EDS analysis results.

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
