# Peer review of "Comparison of Hot Deformation Behavior Characteristics Between As-Cast and Extruded Al-Zn-Mg-Cu (7075) Aluminum Alloys with a Similar Grain Size"

_materials, 2019, doi:10.3390/ma12233807_

Round 1

Reviewer 1 Report

This document presents a physical metallurgy approach to study the thermomechanical behaviour of Al-Zn4 Mg-Cu (7075) aluminium alloys during the hot deformation. The microstructural evolution of alloy is evaluated via EBSD analysis, which readily characterised the details of the dynamic recrystallization occurred in the microstructure. Furthermore, the mechanical properties of the material in the relationship with the microstructural alteration has been investigated.

Although this work achieves its main aim, to be considered as strong research for publishing, it is recommended to consider the next comments.

The relations between the PLB behaviour, equation 3, and presented results need to be elaborated.

Moreover, equations 1 and 2 explain the relationship between flow stress, strain rate and temperature. How you justify the phenomenon explain in equations 1-3 with the results of Figure 4a, in correlation with the microscopic observations.

To build a robust theory, this reviewer suggests the authors extend the discussion to clarify the alteration of the physical phenomenon introduced in section 2.2 during the hot deformation. This approach also can be extended to the power dissipation through the plastic deformation.

Author Response

: I appreciate these suggestions.

>>>>>>To explain the correlation between PLB and flow instability, Figure 6 was newly provided and to show the correlation between PLB and microstructure, Figure 11 was newly provided in the revised paper.

>>>>>>Revised as “ In Figure 4(e), the n1 values of the as-cast and extruded alloys are close to 5 at small Z values where power-law creep dominates plastic flow and then increases to be larger than 7 at large Z values. This result indicates that the rate-controlling deformation mechanisms of the as-cast and extruded alloys are identical and are lattice-diffusion controlled dislocation climb creep at low strain rates and high temperatures and PLB at high strain rates and low temperatures.”

Reviewer 2 Report

This is good work, well presented, that should capture the interest of many readers.  The communication is acceptable; although the English could be improved.  It is suggested that you have an English-as-a-first-language (ESL) person versed in the appropriate science and technology do a critical edit to improve the English.

The location of the Materials and Methods section is not in the normal position for a scientific paper (unless this is a format guideline set by the journal editors), and should be moved to follow the Introduction.  To this section you should add comments on how the temperature was measured and maintained.

Figures 2(a) and 2(b) do not show two different magnifications (see lines 82-83).  Am I not interpreting this comment correctly?

The evidence for the Al7Cu2Fe phase is weak.  Please provide more evidence or improve your argument for this statement (line 89).

Caption for Fig. 3 - describe meaning of open and solid symbols.

Fig. 5 - describe meaning of the color code used and its relationship to contour lines.

Fig. 7 - error bars should be used on data points or a generic error bar shown to give the reader an idea of data scatter and the uncertainty of the measurements.

Conclusions section - it is strongly suggested that an additional conclusion be added that generalizes the findings to other precipitation hardened Al alloys.  This will increase the impact of the work and capture additional reader interest.  To further improve impact, this generalization could be added to the Abstract.

Author Response

Reviewer 2: This is good work, well presented, that should capture the interest of many readers.  The communication is acceptable; although the English could be improved.  It is suggested that you have an English-as-a-first-language (ESL) person versed in the appropriate science and technology do a critical edit to improve the English.

>>>>>>English was polished with aid of English language service (American journal experts)

The location of the Materials and Methods section is not in the normal position for a scientific paper (unless this is a format guideline set by the journal editors), and should be moved to follow the Introduction.  To this section you should add comments on how the temperature was measured and maintained.

: Materials and Methods section was moved to the proper place.

Figures 2(a) and 2(b) do not show two different magnifications (see lines 82-83).  Am I not interpreting this comment correctly?

: The caption was corrected as “Figure 2. The SEM micrographs of the (a) as-cast and (b) extruded alloys.”

The evidence for the Al7Cu2Fe phase is weak.  Please provide more evidence or improve your argument for this statement (line 89).

: Revised as “The XRD curves and the EDS analysis results for the as-cast and extruded alloys shown in Figure 2(c) suggest that the phases segregated at the dendritic interfaces and grain boundaries in the as-cast microstructure, and the particles dispersed in the extruded microstructure are MgZn2, Mg2Si, Al2CuMg, Al7Cu2Fe and Al2Mg3Zn3.”

Caption for Fig. 3 - describe meaning of open and solid symbols.

: The meaning was described.

Fig. 5 - describe meaning of the color code used and its relationship to contour lines.

: The meaning was described.

Fig. 7 - error bars should be used on data points or a generic error bar shown to give the reader an idea of data scatter and the uncertainty of the measurements.

: The error bars were added in Figures 8(a) and (b).

Conclusions section - it is strongly suggested that an additional conclusion be added that generalizes the findings to other precipitation hardened Al alloys.  This will increase the impact of the work and capture additional reader interest.  To further improve impact, this generalization could be added to the Abstract.

: As suggested, the generalized words were added in the abstract and conclusion parts.

Reviewer 3 Report

The present manuscript is devoted to the comparison of the hot deformation behaviors and processing maps of as-cast and extruded 7075 alloys. The obtained results demonstrate high practical significance due to the possibility omitting of the extrusion step for hot working. The paper is well written and contains a lot of experimental data that are clearly discussed. However, there are some comments to pay attention to them.

1) Of course, the Materials and Methods section should be presented after Introduction and before Results.

2)  On page 2 in line 82 the authors have written: "Figures 2(a) and (b) show the SEM micrographs of the as-cast and extruded alloys at two different magnifications." But, Figure 2 has only one magnification. Besides, what is an inset in Fig. 2(b)? There is no description in the text. And what do the Max values 3.32 and 5.71 mean? It should be explained.

3) The caption to Figure 6(f) on page 9 is not correct. Perhaps, 673 K is needed to be written instead of 723 K.

4) On page 10 in line 63, "extruded direction" should be written instead of "compression direction".

5) On the same page the authors have written: "The fractions of dynamically recrystallized grains and HAGBs, which tend to decrease as the temperature decrease and strain rate increase (i.e., Z value increases), are higher in the as-cast alloy compared to the extruded alloy." This sentence is not correct. The DRX grain fraction in as-cast alloy is lower compared to the extruded alloy at low Zener-Hollomon parameter (723 K and 10-3 s-1). In addition, the HAGB fraction in as-cast alloy is higher only for low Z parameters. With an increase in the Z parameter, the HAGB fraction in as-cast alloy becomes lower compared to the extruded alloy. It needs explaining.

6) On page 13 in line 146 "the strain rate range" should be written instead of "the temperature range".

7) On page 14 in line 171 "the five different strain rates" should be written instead of "the four different strain rates".

8) The results of EDS analysis in figure 10 show a very high Si content. The reasons should be explained.

Author Response

Reviewer 3: The present manuscript is devoted to the comparison of the hot deformation behaviors and processing maps of as-cast and extruded 7075 alloys. The obtained results demonstrate high practical significance due to the possibility omitting of the extrusion step for hot working. The paper is well written and contains a lot of experimental data that are clearly discussed. However, there are some comments to pay attention to them.

1) Of course, the Materials and Methods section should be presented after Introduction and before Results.

 : Materials and Methods section was moved to the proper place.

2)  On page 2 in line 82 the authors have written: "Figures 2(a) and (b) show the SEM micrographs of the as-cast and extruded alloys at two different magnifications." But, Figure 2 has only one magnification. Besides, what is an inset in Fig. 2(b)? There is no description in the text. And what do the Max values 3.32 and 5.71 mean? It should be explained.

: All of above mentioned were corrected and explained in the revised paper as: “Figure 1. The IPF maps and image quality maps overlaid with grain boundaries of the (a) as-cast and (b) extruded alloys. The inverse pole figures of the (c) as-cast and (d) extruded alloys. The maximum texture intensity in the inverse pole figures is in the unit of MRD. Low-angle grain boundaries ( ), intermediate-angle grain boundaries ( ) and high-angle grain boundaries ( ), are represented by blue, yellow and red colors, respectively. The inset in (b) shows the color code for the EBSD scans of (a) and (b).”

3) The caption to Figure 6(f) on page 9 is not correct. Perhaps, 673 K is needed to be written instead of 723 K.

: This error was fixed.

4) On page 10 in line 63, "extruded direction" should be written instead of "compression direction".

: This was corrected.

5) On the same page the authors have written: "The fractions of dynamically recrystallized grains and HAGBs, which tend to decrease as the temperature decrease and strain rate increase (i.e., Z value increases), are higher in the as-cast alloy compared to the extruded alloy." This sentence is not correct. The DRX grain fraction in as-cast alloy is lower compared to the extruded alloy at low Zener-Hollomon parameter (723 K and 10-3 s-1). In addition, the HAGB fraction in as-cast alloy is higher only for low Z parameters. With an increase in the Z parameter, the HAGB fraction in as-cast alloy becomes lower compared to the extruded alloy. It needs explaining.

: I am very sorry for this uncarefully stated description. We examined the fraction of DRX grains of the two alloys at 723 K-10-3 s-1 and found that the data of the as-cast and extruded alloys were accidently switched. This was corrected. To explain the lower fraction of HAGBs in the as-cast alloy at large Z values, the following words were added.

REVISED as: “The fractions of DRX grains are higher in the as-cast alloy compared to the extruded alloy. The fractions of HAGBs are higher in the as-cast alloy compared to the extruded alloy at small Z values, but slightly lower in the as-cast alloy at high Z values (above 1011 s-1). The fraction of low-angle grain boundaries (LAGBs) may be higher in the as-cast alloy compared to the extruded alloy at low temperatures due to development of extensive substructure around the segregated phase particles. As temperature increases, however, the fraction of HAGBs increases more rapidly in the as-cast alloy because the number of HAGBs will increase in proportional to the number of LAGBs if LAGBs evolve into HAGBs with progressive increase of misorientation of LAGBs, of which mechanism will be discussed shortly later.

6) On page 13 in line 146 "the strain rate range" should be written instead of "the temperature range".

: This error was fixed.

7) On page 14 in line 171 "the five different strain rates" should be written instead of "the four different strain rates".

: This error was fixed.

8) The results of EDS analysis in figure 10 show a very high Si content. The reasons should be explained.

: The following words were added in the revised paper: “Many of the remaining phases in the as-cast and extruded alloys are mainly made up of undissolved Fe-rich phases, Si-rich phases and Si inclusions. It is known that hard and brittle insoluble intermetallic particles such as Al7Cu2Fe or Si inclusions of the order of 1 to 10 mm in size are frequently found in commercial 7xxx alloys [32].”

Round 2

Reviewer 1 Report

It seems the authors have revised the manuscript according to review comments. The work looks suitable for publishing in this journal.